# Monitored long-range interacting systems: spin-wave theory for quantum trajectories

Zejian Li [1] ✉, Anna Delmonte [2], Xhek Turkeshi [3] & Rosario Fazio [1,4]

Measurement-induced phases exhibit unconventional dynamics as emergent collective phenomena, yet their behavior in tailored interacting systems – crucial for quantum technologies – remains less understood. We develop a systematic toolbox to analyze monitored dynamics in long-range interacting systems, relevant to platforms like trapped ions and Rydberg atoms. Our method extends spin-wave theory to general dynamical generators at the quantum trajectory level, enabling access to a broader class of states than approaches based on density matrices. This allows efficient simulation of large-scale interacting spins and captures nonlinear dynamical features such as entanglement and trajectory correlations. We showcase the versatility of our framework by exploring entanglement phase transitions in a monitored spin system with power-law interactions in one and two dimensions, where the entanglement scaling changes from logarithm to volume law as the interaction range shortens, and by dwelling on how our method mitigates experimental post-selection challenges in detecting monitored quantum phases.

Long-range interacting quantum many-body systems have recently been the focus of intense theoretical and experimental activity[1,2]. Sufficiently non-local interactions between the elementary constituents of a many-body system lead to exotic non-equilibrium phenomena, with compelling signatures in the properties of quantum correlations and entanglement, relaxation dynamics, quantum information scrambling, and ergodicity-breaking properties. Along with their importance in statistical mechanics, long-range quantum systems are central to the rising field of quantum technologies and simulators. Experimental platforms such as trapped ions[3], Bose-Einstein condensates in cavities[4], dipolar[5], polar[6], and Rydberg atoms[7], or driven ultra-cold atomic gases[8] showcase frameworks where the system interactions scale as a power law of the distance.

The dynamics of long-range systems has been investigated in the two limits of unitary and dissipative (Lindbladian) evolutions. In this work, we would like to take a step forward by studying the in-between framework of monitored dynamics. Here, the system evolution is interspersed with quantum measurements, whose outcomes are stochastic. As a result, the system is described by a quantum trajectory conditional to the measurement registry. Averaging the state over the trajectory ensemble recasts a dissipative dynamics, therefore presenting a convenient framework for the study of physical observables in Lindblad dynamics[9–11]. It was more recently realized, in the study of many-body systems, that the quantum trajectory ensemble contains richer information than the mean state, showcased by the several collective phenomena encoded in beyond-average statistical features. The cornerstone examples are monitored quantum phases and measurement-induced phase transitions[12–14], characterized by non-linear functions of the trajectories, such as entanglement probes or trajectory correlation functions. Intensive work gathered insights on the monitored phases in local random quantum circuits[15–20] and non-interacting systems[21,22], while few works have addressed monitored dynamics in long-range interacting systems. References [23,24] studied Clifford circuits with two-qubit gates entangling distant sites with a probability of decaying as a power-law of their distance. The Floquet dynamics of interacting spin systems with measurements and feedback was considered in a model with entanglement and dissipative phase transitions[25], while monitored long-range free fermions have been studied in refs. 26,27. All these results demonstrate that long-range couplings strongly affect

[1]The Abdus Salam International Center for Theoretical Physics, Strada Costiera 11, 34151 Trieste, Italy. [2]SISSA, Via Bonomea 265, I-34136 Trieste, Italy. [3]Institut für Theoretische Physik, Universität zu Köln, Zülpicher Straße 77, 50937 Köln, Germany. [4]Dipartimento di Fisica "E. Pancini", Università di Napoli "Federico II", Monte S. Angelo, I-80126 Napoli, Italy. ✉e-mail: li.zejian@ictp.it

the monitored phases and their transitions: they are relevant in the renormalization group sense.

Experimentally detecting monitored phases, on the other hand, is challenging, and so far limited to few pioneering works on trapped-ions[28] and superconducting platforms[29,30]. This limitation has a fundamental origin known as the post-selection problem. Observing beyond-average moments of the quantum trajectories requires reproducing the same sequence of measurement outcomes multiple times to collect enough statistics and obtain trajectory-average values. This task is formidable, as the probability of reproducing the same trajectory is exponentially small in system size and time scale.

The post-selection barrier is in general ineludible for the study of generic many-body monitored systems, and the quest for methods that can mitigate it is an active research line. For instance, the presence of feedback dynamics may be designed to imprint the measurement-induced transition into the density matrix[31,32], albeit in general leads to separate types of phase transitions[33–38]. A complementary path is available for systems that can be classically simulated. When the post-processing is efficient and faithful, it allows to elude the post-selection[39–45]. There are special cases in which the system itself is designed to be immune to post-selection. In[46] some of us showed that there is a class of infinite-range spin systems where monitored many-body dynamics can be efficiently realized with a post-selection overhead scaling, at most, as a power of the system size. It was argued that this fortunate case was not specific to that model, but it applies to a broad class of monitored systems with an underlying semi-classical dynamics.

A central target of this work is supporting the above claims for sufficiently long-range interacting spin systems. This goal is achieved by

- developing a systematic semi-classical expansion for quantum trajectories that is particularly suited for long-range systems,
- showing that, in this regime, the post-selection barrier is avoidable, paving the way for future experiments. Trapped ions, among others, belong to the systems for which our method and results apply.

As long-range systems are many-body and interacting by construction, their theoretical study also poses a formidable challenge by itself. One may resort to approximative numerical methods, such as those based on matrix-product states[47–49] and other types of tensor networks[50], representing the state-of-the-art for one-dimensional systems and in low entropy scenarios. These methods, while being generic, struggle in capturing correlations built up in nonequilibrium dynamics (for example in a phase transition) and generally suffer from poor complexity scaling in two or higher spatial dimensions. The method we propose, on the other hand, exploits the simplifications granted by long-range interactions: they typically lead to underlying semi-classical dynamics, allowing quantum excitations and fluctuations to be treated perturbatively on top of a classical model. These observations lead to a class of approximation methods known as the spin-wave theory. Historically, the spin-wave theory was first introduced by Bloch[51] in 1932 for ferromagnetic spin systems in equilibrium. An equivalent formulation of the theory was later given by Holstein and Primakoff[52]. The spin-wave theory has been highly successful in describing equilibrium ground states in a wide range of magnetic materials, showing good accordance with experimental data[53]. More recently, it has been generalized to the time-dependent regime[54,55], capturing non-equilibrium dynamics in closed quantum systems undergoing unitary time evolution governed by a Hamiltonian and in driven-dissipative systems[56] described by a Lindblad master equation.

This work enlarges spin-wave theory to encompass generalized measurements, therefore providing a natural framework for the study of monitored Hamiltonian and Lindbladian long-range systems. We propose the theory of spin-wave quantum trajectories (SWQT) and

explain how it is constructed in the case of a monitored dynamics described by a quantum state diffusion. The method is carefully benchmarked with exactly solvable cases and then demonstrated on both one- and two- dimensional spin lattices and on a spin-boson model, exploring different monitored phases where the entanglement scaling ranges from logarithmic to volume law. Moreover, as our method is designed on the level of single quantum trajectories, it also provides a much more accurate representation of the Lindblad dynamics, allowing to capture non-Gaussian corrections that are by default lost in semiclassical approaches approximating directly on the level of the density matrix[56], offering a broad spectrum of applications. Finally, we discuss how the post-selection problem can be avoided in long-range spin systems by exploiting quantum-classical cross-correlated observables enabled by our spin-wave framework.

## Results
### Monitored long-range spin systems
We consider a system with $N$ spin-$s$ degrees of freedom on a lattice whose sites are labeled by indices $i, j$, that is subjected to continuous monitoring. For concreteness, we consider the case of weak monitoring where the dynamics is governed by the following stochastic master equation (in units where $\hbar = 1$),

$$
\begin{aligned}
\mathrm{d}\hat{\rho} = \mathrm{d}t\mathcal{L}(\hat{\rho}) \\
+ \sum_i \left[ \mathrm{dw}_i^* \left( \hat{L}_i - \langle \hat{L}_i \rangle \right) \hat{\rho} + \mathrm{dw}_i \hat{\rho} \left( \hat{L}_i^\dagger - \langle \hat{L}_i^\dagger \rangle \right) \right],
\end{aligned}
\tag{1}
$$

where $\hat{\rho}$ is the density matrix, $\mathcal{L}$ is the Liouvillian superoperator

$$
\mathcal{L}(\hat{\rho}) \equiv -\mathrm{i}[\hat{H}, \hat{\rho}] + \mathcal{D}(\hat{\rho}),
\tag{2}
$$

$\hat{L}_i$ is the Lindblad operator acting on the $i$-th spin, and $\langle \hat{L}_i \rangle \equiv \mathrm{Tr}[\hat{\rho}\hat{L}_i]$ denotes the (single-trajectory) expectation value. In Eq. (2), $\hat{H}$ is the Hamiltonian generating the coherent dynamics, whose form will be specified later, and $\mathcal{D}$ denotes the superoperator governing the dissipation, that acts on the density matrix as follows,

$$
\mathcal{D}(\hat{\rho}) \equiv \sum_{i,j} f_{ij} \left( \hat{L}_i \hat{\rho} \hat{L}_j^\dagger - \frac{1}{2} \left\{ \hat{L}_j^\dagger \hat{L}_i, \hat{\rho} \right\} \right).
\tag{3}
$$

The positive semidefinite matrix $f_{ij}$ takes into account spatial correlations among the Lindblad operators $\hat{L}_i$. In the stochastic master equation (1), $\mathrm{dw}_i$ is a complex Wiener process satisfying the relations

$$
\begin{aligned}
\overline{\mathrm{dw}_i} = 0, \\
\overline{\mathrm{dw}_i^* \mathrm{dw}_j} = f_{ij}\mathrm{d}t, \quad \overline{\mathrm{dw}_i \mathrm{dw}_j} = 0,
\end{aligned}
\tag{4}
$$

where the $\overline{\bullet}$ notation denotes the ensemble average. Note that in the general case where $f_{ij}$ is nondiagonal, the noises are also spatially correlated.

We consider the dynamics of a pure state $\hat{\rho} = |\psi\rangle\langle\psi|$. Given the purity preservation of the dynamics (1) (cf. Supplementary Note I), the von Neumann entropy

$$
S_E \equiv -\mathrm{Tr}\left[ \hat{\rho}_{\frac{N}{2}} \log \hat{\rho}_{\frac{N}{2}} \right]
\tag{5}
$$

is a good quantifier of entanglement[57]. Here, $\hat{\rho}_{\frac{N}{2}} \equiv \mathrm{Tr}_{\{1, \ldots, \lfloor N/2 \rfloor\}}[|\psi\rangle\langle\psi|]$ denotes the half-system reduced density matrix (in a bipartition where one subsystem contains spins indexed from 1 to $\lfloor N/2 \rfloor$).

Using Eqs. (1), (4), one verifies that the trajectory-average state (which we also denote by $\hat{\rho}$ when there is no confusion with the single-trajectory state) evolves deterministically according to the Lindblad

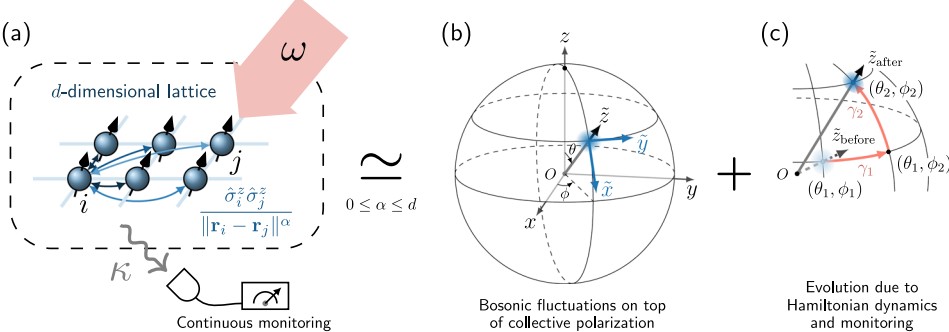

**Fig. 1 | Sketch of a long-range spin model and the representation of its monitored dynamics in the spin-wave approximations. a** The spins on a $d$-dimensional lattice are collectively driven (at amplitude $\omega$) and interact via $z-z$ coupling, whose strength decays as a power law of the distance (with exponent $\alpha$). The system is subjected to a collective dissipation (at rate $\kappa$) and continuously monitored. **b** In the infinite-range or long-range regime ($0 \le \alpha \le d$), the state of the system can be represented in the spin-wave approximations, where spin fluctuations around the collective polarization are bosonized. Here, $Oxyz$ depicts the lab frame and $O\tilde{x}\tilde{y}\tilde{z}$ is the instantaneous frame where $\tilde{z}$ aligns with the collective spin polarization. **c** The evolution of the state under both the Hamiltonian dynamics and the continuous monitoring. In the proposed method, the angles $(\theta, \phi)$ of the instantaneous frame $O\tilde{x}\tilde{y}\tilde{z}$ are updated to match the collective spin direction after its evolution in each infinitesimal time step. The dashed (solid) straight arrow represents the $\tilde{z}$ axis before (after) the re-alignment step and the curved arrows represent the paths $\gamma_1$ and $\gamma_2$ as defined in Eq. (25) for evaluating the integral of Eq. (24).

master equation

$$\frac{\mathrm{d}}{\mathrm{d}t}\hat{\rho} = \mathcal{L}(\hat{\rho}). \tag{6}$$

As shown in Supplementary Note I, Eq. (1) is an unraveling of the master Eq. (6) and describes the quantum state diffusion process subjected to a heterodyne-detection monitoring scheme[10,58].

While the formulation of our spin-wave method will not depend on the particular form of the model, we generally require $\hat{H}$ and $\mathcal{D}$ to be long-range in order for the method to yield an accurate approximation to the exact solution. For concreteness, we study a prototypical driven-dissipative spin model with spatially extended interaction whose strength decays as a power-law according to the distance, as sketched in Fig. 1a. The model is defined on a $d$-dimensional periodic lattice of $N = L^d$ spins at positions $\mathbf{r}_i \equiv (r_i^{(1)}, \ldots, r_i^{(d)})$ where $r_i^{(p)} = 1, \ldots, L$. The Hamiltonian is

$$\hat{H} = \omega \hat{S}^x + \frac{2sJ}{\mathcal{N}} \sum_{i,j} \frac{\hat{\sigma}_i^z \hat{\sigma}_j^z}{\| \mathbf{r}_i - \mathbf{r}_j \|^\alpha}, \tag{7}$$

where $\omega$ is the amplitude of a collective drive and $J$ is the interaction strength. The factor $2s$ ensures that the mean-field theory (see Supplementary Note II) is $s$ – independent, which can be ignored for spin-half ($s = 1/2$) systems. Here, we denote total spin operators with the capital $\hat{S}^\mu$, $\mu \in \{x, y, z\}$ without the site index subscript $\hat{S}^\mu \equiv \sum_i \hat{s}_i^\mu$, while the total spin number is denoted by $S \equiv Ns$. The lower-case notation $\hat{s}_i^\mu$ refers to the spin operator of the $i$ – th site, which satisfies the standard $\mathfrak{su}(2)$ algebra $[\hat{s}_i^\mu, \hat{s}_j^\nu] = \mathrm{i}\delta_{ij}\epsilon^{\mu\nu\gamma}\hat{s}^\gamma$. We denote the normalized spin operators with $\hat{\sigma}_i^\mu \equiv \hat{s}_i^\mu/s$, which reduce to the standard Pauli matrices in the case of spin-half. The distance on the periodic lattice is $\| \mathbf{r}_i - \mathbf{r}_j \| \equiv \sqrt{\sum_{p=1}^d \min(|r_i^{(p)} - r_j^{(p)}|, L - |r_i^{(p)} - r_j^{(p)}|)^2}$, and in the case of $i = j$ we adopt the convention of $\|\mathbf{r}_i - \mathbf{r}_j\| = \infty$ such that there is no on-site self-interaction for finite $\alpha$. The Kac normalization[59] $\mathcal{N} \equiv \frac{1}{N} \sum_{i,j} \| \mathbf{r}_i - \mathbf{r}_j \|^{-\alpha}$, ensures a well-defined thermodynamic limit for the interaction Hamiltonian. The power $\alpha$ determines the range of the interaction. In particular, the case of $\alpha = 0$ describes an infinite-range model with permutation invariance, and the opposite limit $\alpha \to \infty$ corresponds to an Ising model with nearest-neighbor interaction. The long-range regime corresponds to the case where $0 < \alpha \le d$[60].

In addition to the unitary dynamics governed by the Hamiltonian, we subject the spin chain to a collective (infinite-range) decay, a case of experimental relevance[8,61], resulting in the following Lindblad master equation,

$$\frac{\mathrm{d}\hat{\rho}}{\mathrm{d}t} = -\mathrm{i}[\hat{H}, \hat{\rho}] + \frac{\kappa}{S}\left(\hat{S}^- \hat{\rho} \hat{S}^+ - \frac{1}{2}\left\{\hat{S}^+ \hat{S}^-, \hat{\rho}\right\}\right), \tag{8}$$

with $S$ serving the role of Kac normalization for the dissipator, $\kappa$ is the dissipation strength, and $\hat{S}^\pm \equiv \hat{S}^x \pm \mathrm{i}\hat{S}^y$ are the collective spin raising and lowering operators. Note that this collective dissipator corresponds to the choice of $f_{ij} = \kappa/S = \text{const}$ and $\hat{L}_i = \hat{s}_i^- = \hat{s}_i^x - \mathrm{i}\hat{s}_i^y$ in terms of our generic notation in Eq. (3), which also fixes the stochastic unraveling via Eq. (1).

The quantum state of the spin system lives in a $2^N$-dimensional Hilbert space, making the exact solution intractable in practice for thermodynamically large $N$. In the following section, we elaborate the theory of spin-wave quantum trajectories, allowing us to overcome the above limitations and tackle the problem in the thermodynamic limit.

## Spin-wave theory along quantum trajectories

In this section, we present the framework of spin-wave quantum trajectories (SWQT), that is the main result of our work. This formalism serves as a semi-classical method for solving quantum trajectories of generic out-of-equilibrium dissipative/monitored spin systems with sufficiently long-range interactions whose average dynamics can be described by a Lindblad master equation. The resolution of single quantum trajectories enables us to probe non-linear correlations, encoded in the entanglement and other non-linear functions of the state[62].

The central assumption is that the system admits a strong collective spin polarization, on top of which spin-wave excitations are bosonized via a Holstein-Primakoff expansion truncated to the lowest order:

$$\hat{s}_i^{\tilde{z}}(\theta, \phi) = s - \hat{b}_i^\dagger \hat{b}_i,$$
$$\hat{s}_i^{\tilde{x}}(\theta, \phi) \simeq \sqrt{\frac{s}{2}}(\hat{b}_i^\dagger + \hat{b}_i), \tag{9}$$
$$\hat{s}_i^{\tilde{y}}(\theta, \phi) \simeq \mathrm{i}\sqrt{\frac{s}{2}}(\hat{b}_i^\dagger - \hat{b}_i).$$

Here, the $\hat{s}^{\tilde{\alpha}}$'s are spin operators in the rotated frame $O\tilde{x}\tilde{y}\tilde{z}$ as illustrated in Fig. 1b. They are related to the lab-frame spin operators via $\hat{s}_i^{\tilde{\alpha}}(\theta, \phi) = \hat{U}(\theta, \phi)\hat{s}_i^{\alpha}\hat{U}^{\dagger}(\theta, \phi)$ with $\hat{U}(\theta, \phi) = e^{-i\phi\hat{S}^z}e^{-i\theta\hat{S}^y}$. The $\tilde{z}$ axis of the rotated frame is aligned with the collective (average) polarization of all the spins, which can be met by imposing $\langle\hat{S}^{\tilde{x}}\rangle = \langle\hat{S}^{\tilde{y}}\rangle = 0$. The $\hat{b}_i$'s are bosonic operators, representing spin fluctuations around the collective magnetization, with standard bosonic commutation relations $[\hat{b}_i, \hat{b}_j^{\dagger}] = \delta_{ij}$. This transformation effectively approximates the Bloch sphere with the tangent plane at the north pole of the rotated frame[2], and becomes exact only when $\langle\hat{b}_i^{\dagger}\hat{b}_i\rangle = 0$, i.e., when the system is in a spin coherent state. Thus, the density of bosonic excitations

$$\epsilon \equiv \frac{1}{Ns}\sum_i\langle\hat{b}_i^{\dagger}\hat{b}_i\rangle, \tag{10}$$

also referred to as spin-wave density, serves as a natural control parameter for the approximation[56].

The bosonic modes are then approximated with a Gaussian ansatz[63,64] parametrized by first and second moments. This allows us to uniquely specify the state of the entire system with the following variational parameters:

$$\theta, \phi, \beta_i \equiv \langle\hat{b}_i\rangle, u_{ij} \equiv \langle\hat{\delta}_i\hat{\delta}_j\rangle, v_{ij} \equiv \langle\hat{\delta}_i^{\dagger}\hat{\delta}_j\rangle, \tag{11}$$

where $\hat{\delta}_i \equiv \hat{b}_i - \beta_i$. Therefore, this method requires only $\mathcal{O}(N^2)$ parameters to represent the state in the most general case, which is exponentially efficient as compared to the dimension of the full Hilbert space $2^N$, and the complexity can be further reduced in the presence of additional symmetries.

The stochastic master Eq. (1) then translates into the dynamics of the variational parameters (11). Our algorithm is based on the Euler-Maruyama method[65], where time is discretized into small steps $\delta t$. Initializing the parameters at $t = 0$ such that the rotated frame parametrized by $(\theta, \phi)$ has its $\tilde{z}$ axis aligned with the collective spin (which implies $\sum_i \beta_i = 0$), we proceed stroboscopically by repeating the two following steps until the desired time $t$ is reached:

1. Calculate the infinitesimal increments for the Gaussian parameters $\delta\beta_i$, $\delta u_{ij}$ and $\delta v_{ij}$ using Eq. (1), and update these quantities with the increments. Note that the frame angles $(\theta, \phi)$ are kept constant within this step, which implies that the $\tilde{z}$ axis no longer aligns with the collective spin after the update as $\sum_i \delta\beta_i \neq 0$ in general.
2. Update frame angles $(\theta, \phi)$ such that $\sum_i \beta_i = 0$ in the new rotated frame [see Fig. 1c for a schematic representation of this step]. This can be achieved self-consistently by considering the evolution of the bosonic mode $\hat{b}_i$ generated by the (passive) rotation of the frame alone. Then update the Gaussian parameters $\beta_i$, $u_{ij}$ and $v_{ij}$ accordingly (due to the rotation of the frame). Finally, increase the time by $\delta t$ and start a new iteration.

We detail in Methods the key ingredients for performing the two steps sketched above. Note that as this construction does not depend on the spatial structure of the system, it is straightforward to apply the method to arbitrary dimensions and to adapt to different lattice geometries, without comprising its computational efficiency. Moreover, we would like to stress that the framework constructed here is fully generic, as it suffices to systematically apply the bosonization and the Gaussian approximation to obtain the equations of motion. Therefore, it can be straightforwardly extended to other types of systems beyond those we considered. An example of its generalization to spin-boson

systems is presented in Supplementary Note V together with some illustrative numerical results (cf. Supplementary Fig. 10).

Finally, let us remark that since our approximations are formulated at the level of single trajectories and not the average dynamics, our variational ansatz allows us to reach a larger class of density matrices[64] compared to previous methods working on the level of the density matrix[56]. Indeed, the mixture of Gaussian states, such as those describing single trajectories, is a non-Gaussian state in general, hence encodes non-trivial correlations between the spin-waves. This fact has a fundamental operational consequence: spin-wave quantum trajectories provide more accurate representations of the purely Lindblad dynamics, making the SWQT framework compelling also for the study of the dissipative dynamics of hermitian operators.

## Entanglement phase transition in the power-law spin model

In this section, we apply the presented framework of spin-wave quantum trajectories to study the power-law interacting spin model introduced above. We focus on $s = 1/2$ and fix the interaction strength at $J = 0.1\kappa$ for the rest of the section.

We now test the validity of the approximations by evaluating both linear (e.g., observables) and non-linear (e.g., entanglement entropy) functions of the state. In order to benchmark the method, we first revisit the known case of $\alpha = 0$, where the $z - z$ interaction is all-to-all with no spatial resolution. In this regime, the system can be effectively represented as a single spin $S = N/2$, allowing us to benchmark the spin-wave method against the exact simulation of the dynamics in the Dicke basis.

To put the spin-wave quantum trajectories in fair comparison with the exact ones, we integrate the exact stochastic master Eq. (1) using the Euler-Maruyama method, adopting the same time step and the same noise realization as used in the spin-wave calculation. An example of a single-trajectory benchmark for the spin-wave method is shown in Fig. 2a, b, where we compare trajectories for both the magnetization $\langle\hat{S}^{\tilde{z}}\rangle$ and the half-system entanglement entropy (cf. Supplementary Note III and refs. 66,67) $S_E$ against exact ones. The considered system has $S = 64$ and $\omega = 1.25\kappa$, which, as predicted by the mean-field theory, corresponds to the time-crystal phase in the thermodynamical limit. The initial state is the fully polarized Dicke state (all spins pointing up) for both simulations. Surprisingly, the trajectories obtained with the spin-wave method faithfully reproduce the dynamics of both quantities throughout most of the evolution. This suggests that the proposed method can be used as an approximation to resolve dynamics on the level of single trajectories.

We also evaluate the performance of the spin-wave method on trajectory-averaged quantities. Fig. 2c shows the time-evolution of the spin expectations $\overline{\langle\hat{S}^{x,y,z}\rangle}$ obtained by averaging spin-wave quantum trajectories. These are then benchmarked against the exact solution of the Lindblad master Eq. (8). As expected from the single-trajectory performance, the average dynamics is accurately reproduced. Notably, the decaying oscillations of the magnetization are due to the finite size effect and therefore require corrections beyond the mean-field level to capture. It is interesting to note that the length of the averaged collective spin vector $\overline{\langle\hat{S}^{\alpha}\rangle}$ is only a fraction of the maximal value $S$, which implies the that the average density matrix, of highly mixed nature, is far from a state that can be well approximated by the spin-wave approximations. As a consequence, a deterministic version of the spin-wave theory with the same approximations applied to the averaged state, e.g., the one proposed in ref. 56, will completely fail to capture the dynamics in this regime, cf. Supplementary Note IV B and Supplementary Fig. 2. In Fig. 2d, we compare the trajectory-averaged entanglement entropy with that calculated from exact trajectories, which shows a good agreement again. We also report the average spin-

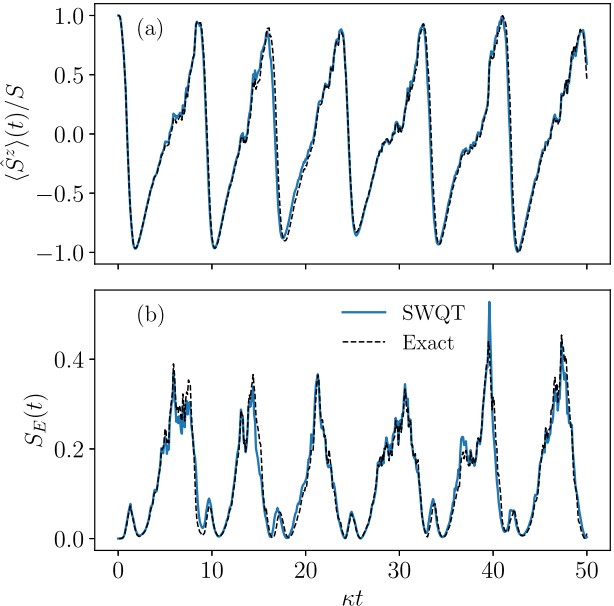

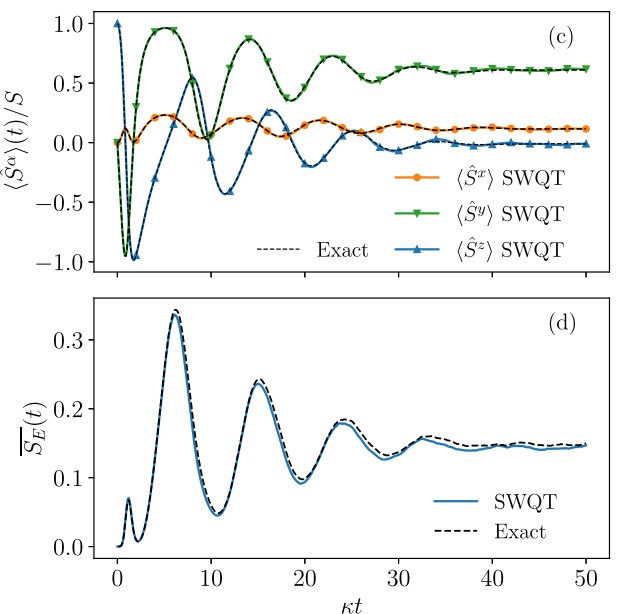

**Fig. 2 | Benchmark of the spin-wave quantum trajectory (SWQT) method against the exact solution for the time-evolution of the following quantities.**

**a** The magnetization $\langle \hat{S}^z \rangle$ along a single trajectory; (**b**) the half-system entanglement entropy $S_E$ along the same trajectory as in (**a**); (**c**) trajectory-average of the collective spin vector $\overline{\langle \hat{S}^\alpha \rangle}$; (**d**) trajectory-average of the half-system entanglement

entropy $\overline{S_E}$. In single-trajectory benchmarks [(**a**) and (**b**)], the same noise realization is adopted by both the spin-wave quantum trajectory and the exact integration of the stochastic master equation. The trajectory-average benchmarks [(**c**) and (**d**)] are performed over 4000 trajectories. Parameters: $\omega = 1.25\kappa$, $J = 0.1\kappa$, $S = 64$, $\kappa\delta t = 10^{-4}$.

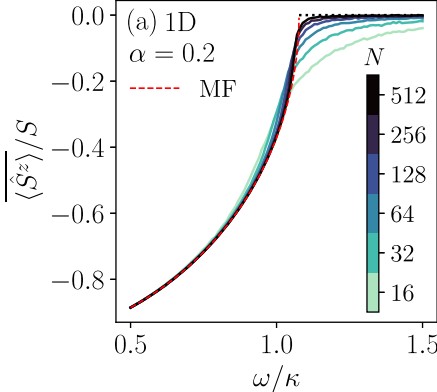

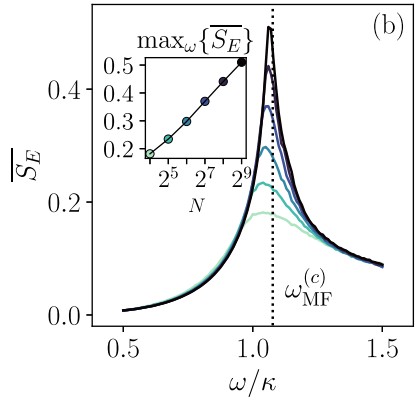

**Fig. 3 | Results for the power-law interacting model in the long-range regime ($\alpha = 0.2$, $J = 0.1\kappa$) of different system sizes (colorbar shared across panels).** The dashed line marks the mean-field (MF) solution. **a** Steady-state average $z$ magnetization as a function of the drive $\omega$. **b** Steady-state of the trajectory-averaged half-

chain entanglement entropy as a function of $\omega$. The vertical dotted line marks the critical point predicted by the mean-field theory $\omega^{(c)}_{\mathrm{MF}} \simeq 1.077\kappa$. Inset: scaling of the maximum entropy versus $N$ in linear-log scale.

wave density to be $\bar{\epsilon} \lesssim 10^{-2}$ throughout the time evolution (see Supplementary Note IV A for a detailed discussion and Supplementary Fig. 1 for additional benchmarks).

To investigate the effect of a finitely long-range interaction and the entanglement dynamics in this regime, we first apply the spin-wave method to the model in one dimension (1D) with $\alpha = 0.2$. Figure 3a shows the steady-state expectation of the collective magnetization $\overline{\langle \hat{S}^z \rangle}$ as a function of the drive amplitude $\omega$ and the system size $N$. As we approach the thermodynamic limit by increasing $N$, the spin-wave solution converges towards the mean-field prediction and a continuous transition emerges separating the normal phase with $\overline{\langle \hat{S}^z \rangle} \neq 0$ and the time-crystal phase with $\overline{\langle \hat{S}^z \rangle} = 0$. Figure 3b shows the behavior of the long-time-averaged half-chain entanglement entropy across this dissipative phase transition. The entanglement develops a

logarithmic divergence as a function of $N$ at the critical point (as implied by the finite-size scaling analysis of the maximum value of the entanglement entropy), while it appears to be $N$-independent for drive values deep inside both phases. This suggests the emergence of an entanglement phase transition separating two area-law phases. We also report (cf. Supplementary Note IV C and Supplementary Fig. 3) that the steady-state spin-wave density $\bar{\epsilon}$ is suppressed with increasing $N$, implying the asymptotic exactness of our results. Due to the sufficiently long-range interaction ($\alpha = 0.2$), these results bear similar features to the infinite-range model ($\alpha = 0$). For larger values of $\alpha$, the entanglement dynamics can be dramatically modified, while the dissipative transition witnessed by the magnetization remains however qualitatively similar since the transition is driven by the infinite-range dissipation and not the long-range coherent interaction. We focus therefore on the entanglement in the discussion that follows. As shown in Fig. 4a, the maximum entanglement for $\alpha = 1.0$ in

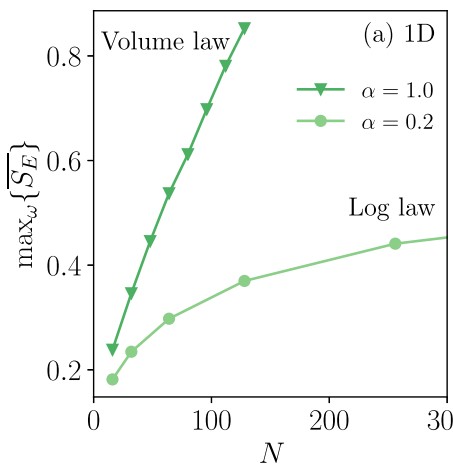
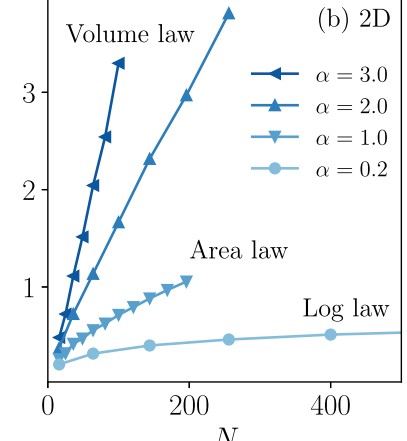

**Fig. 4 | Results on the entanglement scaling in the power-law spin model.** Maximum steady-state entanglement as a function of system size $N$ (total number of spins) for different interaction range $\alpha$ in (**a**) 1D and (**b**) 2D. Different scaling behavior can be observed (see annotation). Note that the horizontal axes are truncated for better visualization and the maximum number of spins we report is $N = 1024$ in 2D (see Supplementary Fig. 5). We report that the value of the drive $\omega$ maximizing the entanglement is asymptotically independent of $N$ (cf. Supplementary Note IV E and Supplementary Fig. 8).

1D exhibits a volume-law scaling in contrast to the log scaling for $\alpha = 0.2$ discussed above. (See Supplementary Note IV D and Supplementary Fig. 4 for additional results in this regime.)

We also perform simulations in 2D (for up to $32 \times 32 = 1024$ long-range interacting spins) to study the entanglement scaling as a function of $\alpha$, as shown in Fig. 4b for a square lattice of size $N = L^2$. As we shorten the interaction range by increasing $\alpha$, the entanglement scaling changes from the log law to area law ($\propto L$) and volume law ($\propto L^2$), which suggests the presence of a phase transition driven by $\alpha$ between the different entanglement scaling behavior. We report (cf. Supplementary Note IV E and Supplementary Figs. 6, 7) that even in the volume-law phase under moderately short-range settings ($\alpha = 2.0$ and $\alpha = 3.0$), the spin-wave density remains small at $\bar{\epsilon} \lesssim 0.1$. On the other hand, the increase of the spin-wave density with higher values of $\alpha$ suggests the eventual breakdown of the spin-wave approximations in the presence of sufficiently short-range interactions.

**Experimental observability of monitored phases boosted by spin-wave quantum trajectories**

The SWQT framework enables operational advantage in the experimental detection of monitored phases of long-range interacting systems. Compared to ref. 46, where the post-selection overhead is mitigated by the permutation symmetry of the setup, this section described a method based on quantum-classical correlations[42,43]. This allows to include permutation symmetry-breaking terms, such as local measurements and power-law decaying interactions, of cornerstone importance for current quantum platforms[3–7,68].

In view of the experimental implementations, for example on digital quantum simulators with discrete-time dynamics, we consider a discretized version of the monitored dynamics, where the noise is binned and approximated with a binary random variable, realized via a set of ancilla qubits in the weak measurement formalism (see Methods for a detailed derivation). We consider an experiment with $\mathcal{M} \gg 1$ weak measurement steps (i.e. projective measurements on the ancilla) and set the measurement outcome history **m**. Suppose we perform a final projective measurement of a system operator $\hat{O}$ over $\hat{\rho}(\mathbf{m})$, i.e. the state conditioned on the measurement history **m**. This measurement is disruptive and will collapse $\hat{\rho}$ onto the eigenspace of the measurement outcome $o_\mathbf{m}$. For example, in the case where the observable is chosen to be $\hat{O} = \hat{S}^z$, the final single-shot measurement outcome $o_\mathbf{m} \in \{-S, \cdots, S\}$ is one of the $2S + 1$ eigenvalues of operator $\hat{S}^z$. Averaging over this quantity recasts the Lindblad

prediction, namely $\overline{o_\mathbf{m}} = \mathrm{Tr}[\hat{\rho}\hat{O}]$ where $\hat{\rho} = \overline{\hat{\rho}(\mathbf{m})}$ is the average state over all possible trajectories **m**. A correlation that is non-linear in the state can be obtained using classical simulations[42]. Fixing the measurement history and position **m** in the classical simulation, we obtain the (classical) estimate of the trajectory-wise expectation value $\langle\hat{O}_\mathbf{m}\rangle_C \equiv \mathrm{Tr}[\hat{\rho}_C(\mathbf{m})\hat{O}]$ with the label $\bullet_C$ denoting the classically computed quantity. For instance, this is obtained in our setup using the solution of Eq. (1) within the spin-wave approximations. We can then cross-correlate the measurement outcome with the classical simulation result to construct the quantum-classical object $o_\mathbf{m}\langle\hat{O}_\mathbf{m}\rangle_C$, which, when averaged over **m** (i.e. over measurement shots),

$$\overline{o_\mathbf{m}\langle\hat{O}_\mathbf{m}\rangle_C} = \overline{\langle\hat{O}_\mathbf{m}\rangle_Q\langle\hat{O}_\mathbf{m}\rangle_C}, \tag{12}$$

gives the quantum-classical correlator, where the label $\bullet_Q$ denotes the quantity evaluated on the actual quantum state in the experiment. In the ideal case where the classical simulation is exact, the quantum and classical quantities should coincide, giving a nonlinear function of the state conditioned on the measurement history. This was recently shown to reproduce the measurement-induced transition in a variety of experiments[30], provided simulating $\langle\hat{O}_\mathbf{m}\rangle_C$ is easy. The SWQT framework therefore is a compelling toolbox for investigating quantum-classical correlations. In particular, when the semi-classical approximation holds, we expect the data to be reliable with errors $O(1/S)$.

To demonstrate this quantum-classical measurement protocol, we revisit the infinite-range ($\alpha = 0$) case of the power-law spin model (fixing again $J = 0.1\kappa$), where the numerically exact solution, that we use to mimic the quantum run, is affordable. The classical quantities are then those given by the SWQT method using the same noise realization **m**. In Fig. 5, we show the steady-state behavior of the quantum-classical correlator $\overline{\langle\hat{S}^z\rangle_Q\langle\hat{S}^z\rangle_C}$ as a function of the drive $\omega$ for different system sizes. Importantly, this quantity successfully captures the signature of the phase transition. This demonstrates that the efficient classical simulation enabled by our spin-wave framework provides access to quantum-classical observables, which can probe nonlinear properties of the state without the post-selection issue.

## Discussion

In this work, we have proposed a stochastic spin-wave theory along quantum trajectories of monitored long-range spin systems. Our

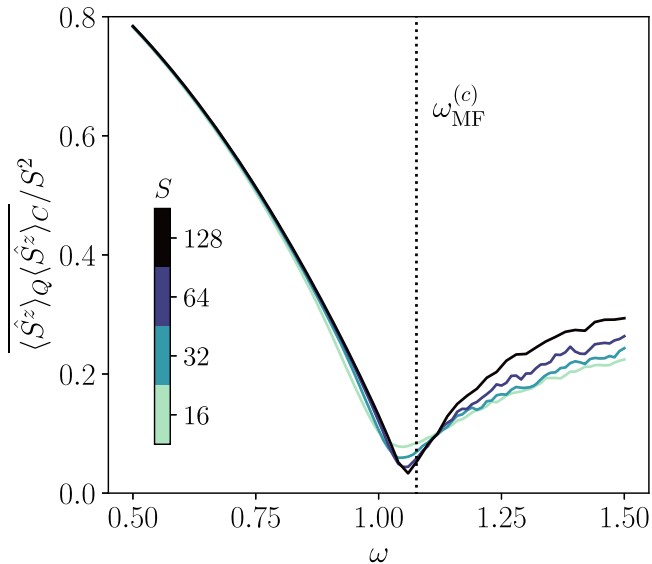

**Fig. 5 | Steady-state trajectory-averaged quantum-classical correlator** $\overline{\langle \hat{S}^z \rangle_Q \langle \hat{S}^z \rangle}_C$ **as a function of the drive $\omega$ for different total spin numbers $S$ (see legend) for the infinite-range model ($\alpha = 0$ and $J = 0.1\kappa$).** The vertical dotted line marks the critical point predicted by the mean-field theory $\omega_{\mathrm{MF}}^{(c)} \simeq 1.077\kappa$. The steady-state value is computed as the average from $\kappa t = 50$ to $\kappa t = 100$, i.e. over the tail of the dynamics.

results illustrate several features of the framework that can be summarized as follows:

- The methodology developed here has a wide applicability not only in the study of entanglement phase transitions but can also be exploited to calculate any other quantity, linear and non-linear in the quantum state along the dynamics either on the level of trajectories or on the Lindblad level, by taking trajectory averages.
- The proposed method works equally well in any dimension, namely its complexity is always polynomial in the total number of qubits. This overcomes the typical limitation of tensor network methods in two or higher dimensions, including volume-law scaling phases.
- When the semi-classical approximations are valid – including the relevant case of sufficiently long-range interactions, the efficient classical simulation of spin-wave quantum trajectories allows to probe nonlinear quantities of the state via quantum-classical cross-correlated observables, which does not suffer from the post-selection problem, thus opening up promising avenues in the experimental detection of monitored phases in long-range systems.

Despite the demonstrated performance of our method, there are several interesting directions for future improvements. For example, one can consider terms in the Holstein-Primakoff expansion beyond the leading order to include more nonlinear effects, which should extend the validity of the theory in the regime of relatively high spin-wave densities. We also expect better overall single-trajectory accuracy with the inclusion of higher-order spin-wave corrections, which is essential in improving the performance of the quantum-classical measurement protocol in avoiding the post-selection problem. On the other hand, different unravelings of the Lindblad master equation can also be explored, including the relevant case of quantum jumps modeling the coupling to photodetectors.

The long-range spin model we adopted for the demonstration of the spin-wave theory can be readily realized in current experimental platforms[8,69], and we expect the entanglement phase transition to be readily observable in experiments without the post-selection problem.

## Methods

### Equations of motion for spin-wave quantum trajectories

We provide in this section the technical details for performing the two steps in evolving the variational state ansatz as sketched in Results.

**Infinitesimal increments.** The stochastic master Eq. (1) determines the time evolution of the expectation value of operators. For a time-independent operator $\hat{O}$, the expectation $\langle \hat{O} \rangle = \mathrm{Tr}[\hat{\rho}\hat{O}]$ evolves as follows,

$$d\langle \hat{O} \rangle = dt \left\langle \mathcal{L}^\dagger(\hat{O}) \right\rangle + \sum_i \left[ dw_i^* \left( \langle \hat{O}\hat{L}_i \rangle - \langle \hat{O} \rangle \langle \hat{L}_i \rangle \right) + dw_i \left( \langle \hat{L}_i^\dagger \hat{O} \rangle - \langle \hat{L}_i^\dagger \rangle \langle \hat{O} \rangle \right) \right], \quad (13)$$

where $\mathcal{L}^\dagger$ is the adjoint Liouvillian:

$$\mathcal{L}^\dagger(\hat{O}) \equiv i[\hat{H}, \hat{O}] + \sum_{i,j} f_{ij} \left( \hat{L}_j^\dagger \hat{O}\hat{L}_i - \frac{1}{2}\left\{ \hat{L}_j^\dagger \hat{L}_i, \hat{O} \right\} \right). \quad (14)$$

The infinitesimal increments for the expectation of the time-independent operator $\hat{b}_i$ can be therefore obtained by setting $\hat{O} = \hat{b}_i$ in Eq. (13):

$$\begin{aligned} d\beta_i &= d\langle \hat{b}_i \rangle \\ &= idt \left\langle [\hat{H}, \hat{b}_i] \right\rangle \\ &+ dt \sum_{j,l} f_{jl} \left\langle \hat{L}_j^\dagger \hat{b}_i \hat{L}_j - \frac{1}{2}\left\{ \hat{L}_l^\dagger \hat{L}_j, \hat{b}_i \right\} \right\rangle \\ &+ \sum_l \left( dw_l^* \langle \hat{\delta}_i \hat{L}_l \rangle + dw_l \langle \hat{L}_l^\dagger \hat{\delta}_i \rangle \right). \end{aligned} \quad (15)$$

For the two-point covariance $u_{ij}$, which is the expectation of the time-dependent observable $\hat{\delta}_i \hat{\delta}_j$, their increments can be obtained using the Ito differentiation rule:

$$\begin{aligned} du_{ij} &= d\langle \hat{\delta}_i \hat{\delta}_j \rangle \\ &= idt \left\langle [\hat{H}, \hat{\delta}_i \hat{\delta}_j] \right\rangle \\ &+ dt \sum_{l,m} f_{lm} \left\langle \hat{L}_m^\dagger \hat{\delta}_i \hat{\delta}_j \hat{L}_l - \frac{1}{2}\left\{ \hat{L}_m^\dagger \hat{L}_l, \hat{\delta}_i \hat{\delta}_j \right\} \right\rangle \\ &+ \sum_l \left[ dw_l^* \left\langle \left( \hat{\delta}_i \hat{\delta}_j - u_{ij} \right)\hat{L}_l \right\rangle + dw_l \left\langle \hat{L}_l^\dagger \left( \hat{\delta}_i \hat{\delta}_j - u_{ij} \right) \right\rangle \right] \\ &- dt \sum_{l,m} \left( f_{lm} \langle \hat{\delta}_i \hat{L}_l \rangle \langle \hat{L}_m^\dagger \hat{\delta}_j \rangle + f_{ml} \langle \hat{\delta}_j \hat{L}_m \rangle \langle \hat{L}_l^\dagger \hat{\delta}_i \rangle \right). \end{aligned} \quad (16)$$

where the last term proportional to $dt$ comes from the time-dependence of $\hat{\delta}_i \hat{\delta}_j$. Similarly, we have for $v_{ij}$,

$$\begin{aligned} dv_{ij} &= d\langle \hat{\delta}_i^\dagger \hat{\delta}_j \rangle \\ &= idt \left\langle [\hat{H}, \hat{\delta}_i^\dagger \hat{\delta}_j] \right\rangle \\ &+ dt \sum_{l,m} f_{lm} \left\langle \hat{L}_m^\dagger \hat{\delta}_i^\dagger \hat{\delta}_j \hat{L}_l - \frac{1}{2}\left\{ \hat{L}_m^\dagger \hat{L}_l, \hat{\delta}_i^\dagger \hat{\delta}_j \right\} \right\rangle \\ &+ \sum_l \left[ dw_l^* \left\langle \left( \hat{\delta}_i^\dagger \hat{\delta}_j - v_{ij} \right)\hat{L}_l \right\rangle + dw_l \left\langle \hat{L}_l^\dagger \left( \hat{\delta}_i^\dagger \hat{\delta}_j - v_{ij} \right) \right\rangle \right] \\ &- dt \sum_{l,m} \left( f_{lm} \langle \hat{\delta}_i^\dagger \hat{L}_l \rangle \langle \hat{L}_m^\dagger \hat{\delta}_j \rangle + f_{ml} \langle \hat{\delta}_j \hat{L}_m \rangle \langle \hat{L}_l^\dagger \hat{\delta}_i^\dagger \rangle \right). \end{aligned} \quad (17)$$

In the equations above, the Hamiltonian $\hat{H}$ and the dissipation operators $\hat{L}_i$ should be expressed in terms of the bosonic operators $\hat{b}_i$ using the substitution rules defined in Eq. (9). The Gaussian approximation then allows evaluating the expectation value of every term beyond quadratic order in $\hat{b}_i$ in terms of one- and two-point correlators, i.e. $\beta_i$, $u_{ij}$ and $v_{ij}$, thanks to the Wick theorem. The increments $\delta\beta_i$, $\delta u_{ij}$, and $\delta v_{ij}$ are then calculated using the discretized versions of the equations above. This can be achieved with the substitution $dt \rightarrow \delta t$ for a

sufficiently small time step $\delta t$. The noise is approximated with $dw_i \rightarrow \delta w_i = \sqrt{\delta t}(X_i + iY_i)$, where the pair of random vectors $(\mathbf{X}, \mathbf{Y})$ at every time step is drawn from a multivariate real Gaussian distribution with zero mean and the following covariance matrix,

$$\mathbf{K} = \begin{pmatrix} \mathbf{K^{XX}} & \mathbf{K^{XY}} \\ \mathbf{K^{YX}} & \mathbf{K^{YY}} \end{pmatrix} \tag{18}$$

with matrix elements

$$(\mathbf{K^{XX}})_{ij} = (\mathbf{K^{YY}})_{ij} = \frac{1}{2}\,\mathrm{Re}\,f_{ij},$$
$$(\mathbf{K^{XY}})_{ij} = -(\mathbf{K^{YX}})_{ij} = \frac{1}{2}\,\mathrm{Im}\,f_{ij}. \tag{19}$$

We complete this step by updating the Gaussian parameters with the increments obtained following the prescription described above:

$$\beta_i \leftarrow \beta_i + \delta\beta_i,$$
$$u_{ij} \leftarrow u_{ij} + \delta u_{ij}, \tag{20}$$
$$v_{ij} \leftarrow v_{ij} + \delta v_{ij}.$$

As a result of the truncated Holstein-Primakoff expansion at the lowest order, the increments include terms up to first order in $1/S$, which account for finite-size effects in the dynamics as a correction to the mean-field (zeroth order) theory.

**Re-alignment of the frame.** After the infinitesimal evolution of the state, let us update the reference frame such that the $\tilde{z}$ axis aligns with the updated direction of the collective spin. This condition is equivalent to requiring the following quantity to be zero,

$$\underline{\beta} \equiv \frac{1}{N}\sum_{i=1}^{N}\beta_i, \tag{21}$$

as a direct implication of Eq. (9). This can be achieved by moving the frame smoothly along a path $\theta(\tau)$, $\phi(\tau)$ parametrized by some parameter $\tau$. The unitary transformation $\hat{U}(\theta, \phi)$ therefore becomes $\tau$-dependent,

$$\hat{U}(\tau) = \hat{U}(\theta(\tau), \phi(\tau)). \tag{22}$$

The rotation of the frame induces some apparent (fictitious) dynamics on the state, whose generator takes the form of an inertial Hamiltonian:

$$\hat{H}_{\mathrm{RF}} = -i\frac{d\hat{U}}{d\tau}\hat{U}^{\dagger}$$
$$= \sin\theta\frac{d\phi}{d\tau}\hat{S}^{\tilde{x}} - \frac{d\theta}{d\tau}\hat{S}^{\tilde{y}} - \cos\theta\frac{d\phi}{d\tau}\hat{S}^{\tilde{z}}. \tag{23}$$

We then use the bosonization rules in Eq. (9) to substitute the spin operators with bosonic ones to express $\hat{H}_{\mathrm{RF}}$ in terms of $\hat{b}_i$. To find the amount of rotation of the frame to achieve $\underline{\beta} = 0$, let us consider the apparent evolution of the operator $\hat{b}_i$ along the moving frame:

$$\frac{d\hat{b}_i}{d\tau} = i\left[\hat{H}_{\mathrm{RF}}, \hat{b}_i\right]$$
$$= -i\sqrt{\frac{s}{2}}\sin\theta\frac{d\phi}{d\tau} - \sqrt{\frac{s}{2}}\frac{d\theta}{d\tau} - i\cos\theta\frac{d\phi}{d\tau}\hat{b}_i. \tag{24}$$

This equation can be integrated analytically by considering the path on the Bloch sphere from $(\theta_1, \phi_1)$ to $(\theta_2, \phi_2)$ following the two segments

[as illustrated in Fig. 1c]:

$$\gamma_1 : \theta(\tau) = \theta_1, \phi(\tau = 0) = \phi_1, \phi(\tau = 1) = \phi_2;$$
$$\gamma_2 : \phi(\tau) = \phi_2, \theta(\tau = 1) = \theta_1, \theta(\tau = 2) = \theta_2. \tag{25}$$

The solution is then

$$\hat{b}_i(\theta_2, \phi_2) = \left[\sqrt{\frac{s}{2}}\tan\theta_1 + \hat{b}_i(\theta_1, \phi_1)\right]e^{-i\Delta\phi\cos\theta_1}$$
$$- \sqrt{\frac{s}{2}}\tan\theta_1 - \sqrt{\frac{s}{2}}\Delta\theta, \tag{26}$$

where $\Delta\theta \equiv \theta_2 - \theta_1$ and $\Delta\phi \equiv \phi_2 - \phi_1$. This equation immediately implies the evolution of $\underline{\beta}$, upon taking the expectation of both sides,

$$\underline{\beta}(\theta_2, \phi_2) = \left[\sqrt{\frac{s}{2}}\tan\theta_1 + \underline{\beta}(\theta_1, \phi_1)\right]e^{-i\Delta\phi\cos\theta_1}$$
$$- \sqrt{\frac{s}{2}}\tan\theta_1 - \sqrt{\frac{s}{2}}\Delta\theta. \tag{27}$$

As our objective is to find the amount of rotations $\Delta\theta$ and $\Delta\phi$ starting from $\theta_1 = \theta$ and $\phi_1 = \phi$ such that $\underline{\beta}(\theta + \Delta\theta, \phi + \Delta\phi) = 0$, we simply set the right-hand side of the equation above to zero, giving our final expressions for the angle increments:

$$\Delta\phi = \frac{1}{\cos\theta}\arctan\left(\frac{\mathrm{Im}\,\underline{\beta}}{\sqrt{\frac{s}{2}}\tan\theta + \mathrm{Re}\,\underline{\beta}}\right)$$
$$\Delta\theta = \left(\tan\theta + \sqrt{\frac{2}{s}}\,\mathrm{Re}\,\underline{\beta}\right)\cos(\Delta\phi\cos\theta)$$
$$+ \sqrt{\frac{2}{s}}\,\mathrm{Im}\,\underline{\beta}\sin(\Delta\phi\cos\theta) - \tan\theta. \tag{28}$$

With the angular increments in hand, the first moments $\beta_i$ of the Gaussian ansatz can be updated directly using Eq. (26) with the operator $\hat{b}_i$ replaced by its expectation value $\beta_i$. This equation also implies the evolution of the fluctuation operators $\hat{\delta}_i = \hat{b}_i - \beta_i$, which is simply

$$\hat{\delta}_i(\theta_2, \phi_2) = \hat{\delta}_i(\theta_1, \phi_1)e^{-i\Delta\phi\cos\theta_1}. \tag{29}$$

We obtain, therefore,

$$u_{ij}(\theta_2, \phi_2) = u_{ij}(\theta_1, \phi_1)e^{-2i\Delta\phi\cos\theta_1},$$
$$v_{ij}(\theta_2, \phi_2) = v_{ij}(\theta_1, \phi_1). \tag{30}$$

With this, we complete the full update step, summarized as follows:

$$\theta \leftarrow \theta + \Delta\theta,$$
$$\phi \leftarrow \phi + \Delta\phi,$$
$$\beta_i \leftarrow \left(\sqrt{\frac{s}{2}}\tan\theta + \beta_i\right)e^{-i\Delta\phi\cos\theta}$$
$$- \sqrt{\frac{s}{2}}\tan\theta - \sqrt{\frac{s}{2}}\Delta\theta,$$
$$u_{ij} \leftarrow u_{ij}e^{-2i\Delta\phi\cos\theta},$$
$$v_{ij} \leftarrow v_{ij}. \tag{31}$$

Finally, we increase the (physical) time $t$ to the next step $t \leftarrow t + \delta t$ and we are ready for a new iteration.

Let us briefly recap the operations performed in each time step of the algorithm. We first update the Gaussian parameters $\beta_i$, $u_{ij}$ and $v_{ij}$ according to Eq. (20) using the increments computed from Eqs. (15)–(17). We then perform the re-alignment step which updates the

variational parameters according to Eq. (31) using the angular increments $\Delta\theta$ and $\Delta\phi$ given by Eq. (28), completing the full iteration.

### From continuous to discrete monitored dynamics

We discuss in this section the discretized version of the dynamics in Eq. (1) by fixing $\Delta t$ a (small) time scale of the unitary and measurement steps, which is equivalent to the continuous-time case in the limit of $\Delta t \to 0$ (see also refs. 10,70). As shown in Supplementary Note I, the stochastic master Eq. (1) can be cast in a diagonal form such that the noises $dZ_j$ associated with different Lindblad jump operators are independent. We consider therefore the simplified problem with a single Lindblad jump operator $\hat{L}$, and focus on the dissipative part of the dynamics (fixing $\hat{H} = 0$), namely

$$d\hat{\rho} = dt\mathcal{D}[\hat{L}](\hat{\rho}) + dZ^*\left(\hat{L} - \langle\hat{L}\rangle\right)\hat{\rho} + dZ\hat{\rho}\left(\hat{L}^\dagger - \langle\hat{L}^\dagger\rangle\right), \quad (32)$$

where the dissipator $\mathcal{D}[\hat{L}]$ is defined as

$$\mathcal{D}[\hat{L}](\hat{\rho}) \equiv \hat{L}\hat{\rho}\hat{L}^\dagger - \frac{1}{2}\left\{\hat{L}^\dagger\hat{L}, \hat{\rho}\right\}, \quad (33)$$

and the zero-mean noise $dZ$ satisfies $|dZ|^2 = dt$ and $dZ^2 = 0$. The generalization to the complete problem is then trivial.

We discretize also the range of values $dZ = (dX + idY)/\sqrt{2}$ into the finite binnings $dX \in \{-x_Q, ..., x_Q\}$ and $dY \in \{-y_Q, ..., y_Q\}$ for some parameter $Q$. The problem in Eq. (32) is then the continuous limit of some positive operator-valued measurements (POVM), which can be described using an ancilla $\mathcal{A}$. We fix $\Delta t > 0$ a small time-step and study the evolution as a stochastic quantum circuit. Let us consider the dynamics of the state $\hat{\mathfrak{R}}_{S,\mathcal{A}}$ describing the combined system and ancilla framework. By definition, we require that $\hat{\rho} = \text{tr}_{\mathcal{A}}(\hat{\mathfrak{R}}_{S,\mathcal{A}})$. We note that the choice of ancilla and system interaction is crucial in determining the value of $dZ$. For simplicity and concreteness, we consider the case of $\mathcal{A}$ being a system of qubits, which has immediate implementations in quantum platforms. In particular, we focus on the minimal setup of two qubits per site, each contributing respectively to the real and imaginary parts of the complex noise. This choice will fix a bimodal approximation of the Gaussian binning, namely $Q = 1$. Nevertheless, the argument below can be generalized to more involved ancillas to reproduce larger values of $Q$.

We denote $\mathcal{A}_{1/2}$ the ancilla qubit 1/2. The heterodyne dynamics in Eq. (32) is then generated by the system-ancilla interaction

$$\hat{U}_{S,\mathcal{A}} = \widehat{BS}e^{\sqrt{\Delta t}(\hat{L}\otimes\hat{\sigma}_1^- - \hat{L}^\dagger\otimes\hat{\sigma}_1^+)}, \quad (34)$$
$$\widehat{BS} = e^{\pi(\hat{\sigma}_1^+\otimes\hat{\sigma}_2^- - \hat{\sigma}_1^-\otimes\hat{\sigma}_2^+)/4},$$

where $\hat{L}$ acts on the system, $\hat{\sigma}_{1,2}^\pm$ are the uppering and lowering operators for the ancilla qubit 1 and 2 respectively, and $\widehat{BS}$ is the 50/50 beamsplitter unitary acting only on the ancilla $\mathcal{A}$.

Applying $\hat{U}_{S,\mathcal{A}}$ to $\hat{\mathfrak{R}}_{S,\mathcal{A}}(t) \equiv \hat{\rho}(t) \otimes |00\rangle\langle00|$ (with the convention where $|0\rangle$ denotes the spin-up state) and projecting out $\mathcal{A}_1$ onto the basis $|\pm\rangle \equiv (|0\rangle \pm |1\rangle)/\sqrt{2}$ and $\mathcal{A}_2$ onto $|\tilde{\pm}\rangle \equiv (|0\rangle \tilde{\pm} i|1\rangle)/\sqrt{2}$ we have the four Kraus operators

$$\hat{K}_{\pm,\tilde{\pm}} \equiv \langle\pm|_{\mathcal{A}_1}\langle\tilde{\pm}|_{\mathcal{A}_2}\hat{U}_{S,\mathcal{A}}|0\rangle_{\mathcal{A}_1}|0\rangle_{\mathcal{A}_2}$$
$$= \frac{1}{2}\left(\mathbb{I} \pm e^{\mp\tilde{\pm}i\pi/4}\sqrt{\Delta t}\hat{L} - \frac{1}{2}\Delta t\hat{L}^\dagger\hat{L}\right), \quad (35)$$

where we have kept terms up to first order in $\Delta t$. We note that the measurement information in the Kraus operators is fully encoded in the 4 complex numbers $\pm e^{\mp\tilde{\pm}i\pi/4}$. Their choice, fixing the measurement history, is therefore determined by the POVM $\hat{E}_{\pm,\tilde{\pm}} = \hat{K}_{\pm,\tilde{\pm}}^\dagger\hat{K}_{\pm,\tilde{\pm}}$.

The post-measurement state is

$$\hat{\rho}_{\pm,\tilde{\pm}} = \frac{\hat{K}_{\pm,\tilde{\pm}}\hat{\rho}\hat{K}_{\pm,\tilde{\pm}}^\dagger}{p_{\pm,\tilde{\pm}}}, \quad (36)$$

where the probabilities are given by

$$p_{\pm,\tilde{\pm}} = \text{tr}(\hat{\rho}\hat{E}_{\pm,\tilde{\pm}}). \quad (37)$$

We will now briefly show that Eq. (36) reproduces Eq. (32) for small $\Delta t$, with a binary approximation of the binning. We define two stochastic variables depending on the measurement outcomes $\Delta R_x(\pm, \tilde{\pm}) \equiv \pm\sqrt{\Delta t}$ and $\Delta R_y(\pm, \tilde{\pm}) \equiv \tilde{\pm}\sqrt{\Delta t}$. We then have, at leading order in $\Delta t$, that

$$\overline{\Delta R_x} = \sum_{\pm,\tilde{\pm}} \pm\sqrt{\Delta t}p_{\pm,\tilde{\pm}} = \Delta t\langle\hat{L}^\dagger + \hat{L}\rangle/\sqrt{2},$$
$$\overline{\Delta R_y} = \sum_{\pm,\tilde{\pm}} \tilde{\pm}\sqrt{\Delta t}p_{\pm,\tilde{\pm}} = i\Delta t\langle\hat{L}^\dagger - \hat{L}\rangle/\sqrt{2}. \quad (38)$$

In a similar fashion, we have $\overline{\Delta R_x^2} = \overline{\Delta R_y^2} = \Delta t$ and $\overline{\Delta R_x \Delta R_y} = 0$. Putting all together, and defining the zero-mean stochastic processes $dX \equiv \Delta R_x - \overline{\Delta R_x}$ and $dY = \Delta R_y - \overline{\Delta R_y}$, which satisfy $\overline{dX^2} = \overline{dY^2} = \Delta t$ and $\overline{dXdY} = 0$, we obtain the final expression after simple algebra,

$$\delta\hat{\rho}_{\pm,\tilde{\pm}} = \hat{\rho}_{\pm,\tilde{\pm}} - \hat{\rho}$$
$$= \Delta t\mathcal{D}[\hat{L}]\hat{\rho} + \frac{1}{\sqrt{2}}dX\left(\hat{L}\hat{\rho} + \hat{\rho}\hat{L}^\dagger - \langle\hat{L} + \hat{L}^\dagger\rangle\hat{\rho}\right)$$
$$+ \frac{i}{\sqrt{2}}dY\left(-\hat{L}\hat{\rho} + \hat{\rho}\hat{L}^\dagger + \langle\hat{L} - \hat{L}^\dagger\rangle\hat{\rho}\right), \quad (39)$$

which is equivalent to Eq. (32) when substituting $dZ = (dX + idY)/\sqrt{2}$. We benchmark the validity of this binary approximation in Supplementary Note IV F (cf. Supplementary Fig. 9).

In summary, within the choice of ancilla and system-ancilla interaction, we obtain a 4-valued complex process $dZ$ for each independent Lindblad jump operator $\hat{L}$ at each monitoring step. This allows us to estimate the brute-force post-selection overhead over $M$ discrete timesteps for $\Lambda$ independent Lindblad jump operators as $O(4^{M\Lambda})$. More generally, introducing further ancilla qubits, or enabling qubit interactions, allows one to reach a more refined binning of the Gaussian increment $dZ$. Fixing the binning parameter $Q$, the probability of reproducing the same trajectory is then $O((2Q)^{2M\Lambda})$. This renders the brute-force experimental observation of the monitored phases of generic systems an exponentially hard task.

## Data availability

The data generated in this study has been deposited in the Zenodo public folder[71].

## Code availability

The computer code developed in this study has been deposited in the Code Ocean capsule associated with the article.

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

## Acknowledgements

We would like to thank Marco Schirò, Gerald E. Fux, Procolo Lucignano, Gianluca Passarelli, Silvia Pappalardi, Cristiano Ciuti, and Valentin Heyraud for helpful discussions. This work was supported by PNRR MUR project PE0000023- NQSTI, by the European Union (ERC, RAVE, 101053159). X.T. acknowledges support from DFG under Germany's Excellence Strategy—Cluster of Excellence Matter and Light for Quantum Computing (ML4Q) EXC 2004/1 - 390534769, and DFG Collaborative Research Center (CRC) 183 Project No. 277101999—project B01. Views and opinions expressed are however those of the author(s) only and do not necessarily reflect those of the European Union or the European Research Council. Neither the European Union nor the granting authority can be held responsible for them.

## Author contributions

Z.L., A.D., X.T. and R.F. contributed to the conception and the implementation of the research, to the discussion of the results, and to the writing of the manuscript.

## Competing interests

The authors declare no competing interests.

## Additional information

**Peer review information** : *Nature Communications* thanks the anonymous reviewer(s) for their contribution to the peer review of this work. A peer review file is available.

