## [Transparent Peer Review file · Nature Communications]

Monitored long-range interacting systems: spin-wave theory for quantum trajectories

Corresponding Author: Dr Zejian Li

Version 0:

Reviewer comments:

Reviewer #2

(Remarks to the Author)

In this paper, the authors propose a numerical scheme for monitored spin systems, with the methodology detailed in Protocols 1 and 2 in Section 2.B. Monitored many-body systems have recently attracted considerable attention, leading to significant breakthroughs, such as measurement-induced phase transitions in entanglement entropy and dissipative phase transitions. The trajectory-level dynamics, however, present complexities beyond previous assumptions, revealing nontrivial phase transitions that are not apparent in the density matrix formulation. The authors introduce a protocol to explore trajectory dynamics via spin-wave dynamics, which, in principle, can be scaled to larger systems. Given the inherent challenges in simulating many-body quantum dynamics in extensive systems, this approach could be a valuable contribution to the field. Nevertheless, I have several concerns regarding the purported advantages of the method.

(i) The dynamics appear to be dependent on the specific form of equation (1), which utilizes a one-body creation (or annihilation) operator as the Lindblad operator L . How robust would this method be if alternative dynamics were considered, or if a different Lindblad operator, such as $L = s^\dagger$, were employed?

(ii) Intuitively, the method seems to perform optimally only in cases of low entanglement. If this holds, it would represent a substantial limitation. The authors should assess the accuracy of this approach not only in the low-entanglement regime but also within the volume-law phase.

(iii) What are the primary applications of this method? In low-entanglement scenarios, alternative techniques, such as matrix product operators (MPOs) or matrix product states (MPS) under measurement, also prove viable. In what contexts does this method offer a distinct advantage?

At present, I am unable to recommend this paper for publication, as the benefits of the proposed method remain insufficiently clear.

(Remarks on code availability)

Reviewer #3

(Remarks to the Author)

In this work the authors develop a spin-wave theory for (monitored) quantum trajectories, providing an efficient numerical approach for simulating the dynamics of e.g. magnetization and entanglement, which can then additionally be used in the experimental detection of monitored phases through the use of quantum-classical correlations. The latter avoids the problem of post-selection, since this approach provides access to quantities that are nonlinear in the density matrix. While the developed framework of spin-wave quantum trajectories (SWQT) can in principle be applied to any spin Hamiltonian, it is expected to be particularly effective in systems with long-range interactions, as is demonstrated in this work. The results are convincing and interesting, from both a theoretical and a practical point of view, and the manuscript is very well written. I

expect these results to have direct impact in the field of quantum simulation and in the study of systems with long-range interactions.

There are two main contributions of this manuscript. The first is the development of the spin-wave theory for trajectories, which relies on bosonization through a Holstein-Primakoff expansion combined with a rotating wave transformation. The involved mathematics is relatively straightforward and it is clear that the method should perform well for systems with infinite-range and sufficiently long-range interactions, as demonstrated in the manuscript. The second main contribution is showing how SWQT can be used to theoretically and experimentally detect an entanglement phase transition, which is again convincingly demonstrated. One criticism to this work is that its use to detect an entanglement phase transition in a monitored spin system with long-range interactions is close to some of the authors' previous works (Ref. [48]), where a similar result is shown for a similar system with infinite-range interactions. In the regime considered in this work the dynamics of both systems are expected to be qualitatively similar, such that the success of this approach is unsurprising. The extension from infinite-range to long-range is however a nontrivial result.

While the manuscript easily meets Nature Communication's criteria for the quality of the data and the level of support for the conclusions, I am not fully convinced that the significance of this work meets the journal's criteria. If the authors could make a better case for the application of their method and how the results on the detection of entanglement transitions are conceptually different from those for systems with infinite-range interactions (and their Ref. [48] more specifically) I would consider recommending this work for publication in Nature Communications.

(Remarks on code availability)

Version 1:

Reviewer comments:

Reviewer #2

(Remarks to the Author)

I appreciate that the authors have adequately addressed the referee's comments. While I agree with some of their responses, I find that their discussion on the general applicability of the spin-wave method is somewhat overstated. The authors have provided an appropriate response regarding its applicability, but since the spin-wave method inherently assumes strong spin polarization, it cannot be universally applied. They emphasize its applicability in various cases, including two-dimensional systems, without sufficiently explaining the scenarios in which it fails. This, in turn, weakens the overall persuasiveness of their argument.

Regarding the comparison with existing tools, I find their explanation reasonable, as it suggests a complementary relationship with other methods. However, while the authors claim that the entropy follows a volume law, their numerical calculations do not convincingly demonstrate that the scaling is strictly extensive.

The numerical method they employ is undoubtedly novel, but at the same time, its applicability is likely limited to a certain class of systems. Identifying these limitations should be addressed in future studies. Given the current state of their approach, I am not entirely convinced that it warrants unconditional praise. For these reasons, I do not recommend publication at this stage, but I defer to the editor's final decision

(Remarks on code availability)

Reviewer #3

(Remarks to the Author)

All of my comments have been appropriately addressed by the additional simulations and comments, and I would like to thank the authors for their detailed reply. I am happy to recommend this paper for publication in Nature Communications.

(Remarks on code availability)

REPLY TO REVIEWER #2

Referee: *In this paper, the authors propose a numerical scheme for monitored spin systems, with the methodology detailed in Protocols 1 and 2 in Section 2.B. Monitored many-body systems have recently attracted considerable attention, leading to significant breakthroughs, such as measurement-induced phase transitions in entanglement entropy and dissipative phase transitions. The trajectory-level dynamics, however, present complexities beyond previous assumptions, revealing nontrivial phase transitions that are not apparent in the density matrix formulation. The authors introduce a protocol to explore trajectory dynamics via spin-wave dynamics, which, in principle, can be scaled to larger systems. Given the inherent challenges in simulating many-body quantum dynamics in extensive systems, this approach could be a valuable contribution to the field. Nevertheless, I have several concerns regarding the purported advantages of the method.*

Reply: We would like to thank Reviewer #2 for the detailed and thoughtful report. Here is a point-by-point answer to the questions and comments raised by the Reviewer.

Referee: *The dynamics appear to be dependent on the specific form of equation (1), which utilizes a one-body creation (or annihilation) operator as the Lindblad operator L . How robust would this method be if alternative dynamics were considered, or if a different Lindblad operator, such as $\hat{L} = \hat{s}^\dagger \hat{s}$, were employed?*

Reply: We thank the Reviewer for this insightful question. The dynamics defined by equation (1) is not restricted to one-body operators. As explained in Section II B, our method employs a semi-classical expansion, applicable to *any generic Lindblad operator*.

For instance, consider $\hat{L} = \hat{s}^\dagger \hat{s}$, where $\hat{s} = c_x \hat{s}^x + c_y \hat{s}^y + c_z \hat{s}^z$ in the laboratory frame. In the frame aligned with the classical spin direction, \hat{L} can be expressed as a combination of rotated operators $\hat{s}^{\hat{\alpha}}$, which are then bosonized using Eq. (9). The resulting bosonic operators \hat{b} and \hat{b}^\dagger are substituted back into \hat{L} , enabling the derivation of spin-wave dynamics.

In the equations of motion, moments of \hat{b} beyond quadratic order are handled using Wick's theorem and our Gaussian approximation [see paragraph after Eq. (17)], ensuring the equations remain closed at the Gaussian level.

Since this procedure is independent of the specific form of the Lindblad (or Hamiltonian) operator, the robustness of our method is expected to hold broadly. We also remark that

alternative dynamics (e.g., different unravelings) can be implemented within this framework, as well as other types of systems such as a spin-boson model as demonstrated in Supplementary Note V. Finally, we note that the specific case studied in the manuscript represents a challenging scenario, as existing spin-wave theories for mixed states typically fail here (cf. Supplementary Note IV B).

Referee: *Intuitively, the method seems to perform optimally only in cases of low entanglement. If this holds, it would represent a substantial limitation. The authors should assess the accuracy of this approach not only in the low-entanglement regime but also within the volume-law phase.*

Reply: We thank the Reviewer for raising this important issue, which was not transparently addressed in the previous version of the manuscript. *Our method can capture volume-law phases.*

For instance, the dynamics of our long-range model is known to reach a volume-law entanglement scaling at $\alpha = d$, with d the spatial dimension. In the previous version of the manuscript, we discussed the one-dimensional case with $\alpha = 1$, which exhibits volume-law scaling of entanglement, as shown in Fig. 1(a). This figure, reproduced from the inset of Fig. S4(b) in Supplementary Note IVD, illustrates that despite the volume-law scaling of entanglement, the spin-wave density $\bar{\epsilon}$ remains bounded and relatively small (peaking at approximately 0.1).

To further clarify this behavior, we conducted new simulations for two-dimensional systems. In the revised manuscript, we included a new figure illustrating the two-dimensional case with $\alpha = 2.0$, shown in Fig. 1(b), which we compare with the case $\alpha = 0.2$ that exhibits a logarithmic scaling of entanglement, cf. Fig. 1(c), reproduced from the insets of Figs. S6 and S5 in Supplementary Notes respectively. *En passant*, these additions aim to provide deeper insight into the robustness of our findings across different dimensionalities.

Referee: *What are the primary applications of this method? In low-entanglement scenarios, alternative techniques, such as matrix product operators (MPOs) or matrix product states (MPS) under measurement, also prove viable. In what contexts does this method offer a distinct advantage?*

FIG. 1. Entanglement scaling in different settings: (a) 1D chain with $\alpha = 1.0$ showing a volume-law (results up to $N = 128$ spins); (b) 2D square lattice with $\alpha = 2.0$ also with a volume-scaling (up to $N = 16 \times 16 = 256$ spins); (c) 2D square lattice with $\alpha = 0.2$ with a logarithmic scaling (up to $N = 32 \times 32 = 1024$ spins). In all plots, N refers to the *total* number of spins in the system. Note that the values of the drive ω maximizing the entanglement are asymptotically independent of N in each panel, and we typically observe the maximum to be achieved around $\omega \simeq 1.07\kappa$ in the cases considered here (see Figs. S4, S6, S5 and S8 in Supplementary Notes for full figures.)

Reply: We thank the Referee for this insightful question, which provides an opportunity to further clarify the scope and robustness of the methodology we have developed.

First, we would like to emphasize that our work addresses problems that often fall beyond the reach and are complementary of traditional tensor network approaches. An important example are higher-dimensional problems. While we cannot exclude that clever tensor network implementation can be adapted to tailored two-dimensional systems, several subtleties enter (e.g., order of contractions, increment of bond dimension with long-range interactions).

The key advantage of the monitored spin-wave theory we propose lies in its inherent semiclassical framework. Within this approximation, that is self-consistently controlled by the spin-wave density, the method can describe higher-dimensional systems with classical resources that scale quadratically with the number of qubits, hence is always polynomial in system size [as explained in the paragraph after Eq. (11) in the Main Text]. A concrete example is presented within the volume-law phase presented in Fig. 1 above (cf. also Fig. 2 in the reply to Reviewer #3 below).

The new results, included in the revised manuscript, show a volume-law scaling entan-

glement for up to $16 \times 16 = 256$ spins on a 2D square lattice, which might be challenging for alternative techniques based on matrix-product methods. We also performed a simulation for the 2D system in the long-range regime ($\alpha = 0.2$) in the revised version, as shown in Fig. 1 (c) in this document. Despite the log-law scaling of entanglement (which is sub-volume), we demonstrate results for considerably large sizes (up to $32 \times 32 = 1024$ spins, see full figures in the newly added Supplementary Note IV E). Finally, the application of the method is not limited to the computation of entanglement. Any linear or non-linear function of the state can be obtained, with an error fixed by the spin-wave density.

Referee: *At present, I am unable to recommend this paper for publication, as the benefits of the proposed method remain insufficiently clear.*

Reply: We hope that the new results and clarifications provided in the revised manuscript address the Reviewer's concerns and demonstrate that our work meets the acceptance criteria for Nature Communications.

REPLY TO REVIEWER #3

Referee: *In this work the authors develop a spin-wave theory for (monitored) quantum trajectories, providing an efficient numerical approach for simulating the dynamics of e.g. magnetization and entanglement, which can then additionally be used in the experimental detection of monitored phases through the use of quantum-classical correlations. The latter avoids the problem of post-selection, since this approach provides access to quantities that are nonlinear in the density matrix. While the developed framework of spin-wave quantum trajectories (SWQT) can in principle be applied to any spin Hamiltonian, it is expected to be particularly effective in systems with long-range interactions, as is demonstrated in this work. The results are convincing and interesting, from both a theoretical and a practical point of view, and the manuscript is very well written. I expect these results to have direct impact in the field of quantum simulation and in the study of systems with long-range interactions.*

Reply: We would like to thank Reviewer #3 for the detailed and thoughtful report. Here is a point-by-point answer to the questions and comments raised by the Reviewer.

Referee: *One criticism to this work is that its use to detect an entanglement phase transition in a monitored spin system with long-range interactions is close to some of the authors' previous works (Ref. [48]), where a similar result is shown for a similar system with infinite-range interactions. In the regime considered in this work the dynamics of both systems are expected to be qualitatively similar, such that the success of this approach is unsurprising. The extension from infinite-range to long-range is however a nontrivial result.*

Reply: We thank the Reviewer for raising this comment. We agree that extending from infinite-range to long-range interactions is a nontrivial challenge. In the long-range regime we considered, where $\alpha < 1$, our method demonstrates the capability to explore much larger system sizes and operates effectively in any spatial dimension (in contrast with Ref. [48]).

To further substantiate this, we conducted new simulations in 2D, as presented in Figs. 1 and 2 here, as well as in additional figures included in the revised manuscript. These results illustrate that even for moderately short-range cases (e.g., $\alpha \simeq 2$), the method continues to produce reliable results, enabling us to investigate volume-law entanglement phases. This

underscores the versatility and robustness of the methodology, as also discussed in our response to the next question.

Referee: *While the manuscript easily meets Nature Communication's criteria for the quality of the data and the level of support for the conclusions, I am not fully convinced that the significance of this work meets the journal's criteria. If the authors could make a better case for the application of their method and how the results on the detection of entanglement transitions are conceptually different from those for systems with infinite-range interactions (and their Ref. [48] more specifically) I would consider recommending this work for publication in Nature Communications.*

Reply: We thank the Reviewer for the constructive criticism, which has helped us improve the presentation of our main results and better emphasize the conceptual challenges addressed by our method. A summary of the main conceptual advances provided by our method is presented below:

- **Beyond semiclassical Lindblad theory.** The proposed method has a wide applicability not only in the study of entanglement phase transitions but can also be exploited to calculate any other quantity, linear and non-linear in the quantum state along the dynamics either on the level of trajectories or on the Lindblad level, by taking trajectory averages. Within this context, the method allows to capture of non-Gaussian corrections that are by default lost in approximating directly the Lindbladian of semiclassical systems.
- **Volume-law phases of interacting (2+1)-dimensional monitored systems.** The proposed method works equally well in any dimension, namely its complexity is always quadratic in the total number of qubits. This enables to overcome the typical limitation of tensor network methods in two or higher dimensions, including volume-law scaling phases.

In this revised version, we include simulations of a novel entanglement phase transition as a function of the interaction range α , presented in the new Fig.2 of this reply and added to the manuscript as Fig.4. As the exponent α increases from long-range to short-range interactions on a 2D square lattice, the entanglement scaling transitions

FIG. 2. Maximum steady-state entanglement as a function of system size $N = L^2$ (total number of spins) for different interaction range α . Different scaling behavior (log law, area law and volume law) can be observed for increasing α . As similarly reported in Fig. 1 here, the drive ω maximizing the entanglement is asymptotically independent of N , at around $\omega \simeq 1.07\kappa$.

from logarithmic to area-law ($\propto L$) and eventually to volume-law ($\propto L^2$). This behavior suggests the presence of a phase transition driven by α . This new result underscores both the efficiency and the broad applicability of our method in exploring monitored many-body phases, particularly in challenging regimes such as the volume-law phase in 2D systems, previously tackled only in Clifford circuits. (See also Fig. 1 and the discussion in the reply to Reviewer #2.)

- Post-selection mitigation beyond the infinite-range regime.** A distinctive trait of the methodology developed here is allowing to substantially reduce post-selection overhead in experimental setups, leveraging on conceptually different strategies than the previous works in the infinite-range regime (Ref. [48]). While the latter relies on the fast-saturating entanglement dynamics and brute-force post selection (which is a feature also present in the sufficiently long-range regime, as we discussed in Supplementary Notes IV C and V extending these results to a spin-boson system), our method is based on the single-trajectory simulability offered by the spin-wave approximations, which can be used to construct quantum-classical correlators [Eq. (12) of the main text] for the experimental detection of the entanglement phase transition. Furthermore, as the method also works in regimes with volume-law entanglement (that

no longer presents fast saturating dynamics), the proposed detection method can still be applied.

We hope that with these new results and clarifications, the Reviewer will find our work suitable to meet the acceptance criteria of Nature Communications.

SUMMARY OF CHANGES IN THE MANUSCRIPT

Note that the modified text in the manuscript are **highlighted with magenta color**.

- We have expanded the last sentence of the abstract to emphasize the new results in 2D and volume-law scaling.
- We have added a short discussion on tensor networks at the beginning of the third last paragraph of the introduction and added the associated Refs. [49-52].
- We have modified the second last paragraph of the introduction to highlight our new results and the advantages of the method.
- We have changed the schematic representation of the long-range model in Fig. 1 from 1D to 2D (and modified the caption accordingly) to incorporate the new results.
- We have slightly modified the model description in Sec. II A (originally only for 1D) to a more general definition to include the 2D case considered in the new version.
- We have moved the paragraph originally at the end of Sec. IV A to the second last paragraph of Sec. II B and stressed the generality of the method.
- We have added Fig. 2 shown above (together with its 1D version) to the main text as Fig. 4 and added a short discussion on the different entanglement scaling at the end of Sec. II C 2.
- We have rearranged the first paragraph of the discussion section (Sec. III) based on the reply to the second question of Reviewer #3 to highlight the main features of the proposed framework.
- We have rephrased the last sentence of Supplementary Note IV D.
- We added Supplementary Note IV E on the additional results in 2D.
- We have corrected some minor typos.

REPLY TO REVIEWER #2

Referee: *I appreciate that the authors have adequately addressed the referee's comments. While I agree with some of their responses, I find that their discussion on the general applicability of the spin-wave method is somewhat overstated. The authors have provided an appropriate response regarding its applicability, but since the spin-wave method inherently assumes strong spin polarization, it cannot be universally applied. They emphasize its applicability in various cases, including two-dimensional systems, without sufficiently explaining the scenarios in which it fails. This, in turn, weakens the overall persuasiveness of their argument.*

Reply: We thank the Reviewer #2 for the positive assessment of our previous response and improved manuscript, and thank them for the additional comments.

We agree that the applicability of our spin-wave method relies on the strong spin polarization assumption. However, this is granted by the long-range nature of the systems of interest in this study, which we verified using the spin-wave density as a self-consistent control parameter, as already discussed in the manuscript.

We have added a sentence at the end of Sec. IIC to state this limitation more explicitly: "On the other hand, the increase of the spin-wave density with higher values of α suggests the eventual breakdown of the spin-wave approximations in the presence of sufficiently short-range interactions."

Referee: *Regarding the comparison with existing tools, I find their explanation reasonable, as it suggests a complementary relationship with other methods. However, while the authors claim that the entropy follows a volume law, their numerical calculations do not convincingly demonstrate that the scaling is strictly extensive.*

Reply: We thank the Reviewer for appreciating our explanation. To better support the claim on volume-law scaling, we have added linear-fit trend lines in the Supplementary figures S4 (for 1D), S6 and S7 (for 2D) and report the coefficient of determination of the fits ($R^2 = 0.9980, 0.9991$ and 0.9983 respectively). These results unambiguously identify the scaling is extensive.

Referee: *The numerical method they employ is undoubtedly novel, but at the same time,*

its applicability is likely limited to a certain class of systems. Identifying these limitations should be addressed in future studies. Given the current state of their approach, I am not entirely convinced that it warrants unconditional praise. For these reasons, I do not recommend publication at this stage, but I defer to the editor's final decision

Reply: We thank the Reviewer for appreciating the novelty of our method. As addressed in the first reply and also in the manuscript, this method is tailored for long-range systems, whose limitations can be self-consistently verified by the spin-wave density control parameter. We will be happy to further explore the applicability/limitation of the method in future studies, for example in systems with shorter-range interactions, which falls beyond the scope of the current work.

REPLY TO REVIEWER #3

Referee: *All of my comments have been appropriately addressed by the additional simulations and comments, and I would like to thank the authors for their detailed reply. I am happy to recommend this paper for publication in Nature Communications.*

Reply: We thank the Reviewer #3 again for the positive assessment of our work.

SUMMARY OF CHANGES IN THE MANUSCRIPT

Note that the modified text in the manuscript are **highlighted with magenta color**.

- We have added a sentence at the end of Sec. IIC to address the limitation of the method: “On the other hand, the increase of the spin-wave density with higher values of α suggests the eventual breakdown of the spin-wave approximations in the presence of sufficiently short-range interactions.”
- We have added linear-fit trend lines in the Supplementary figures S4 (for 1D), S6 and S7 (for 2D) and reported the coefficient of determination of the fits.
- We have made the necessary modifications to comply with the formatting guidance. (Not highlighted.)